# Potential Mechanism of Optimal Tillage Layer Structure for Improving Maize Yield and Enhancing Root Growth in Northeast China

Hongbing Zheng [1,2], Ruiping Li [1,2], Pengxiang Sui [1,2], Hao Wang [1,2], Ying Ren [1,2], Ye Yuan [1,2], Shengtao Tian [1,2], Siqi Zhou [1,2], Wuren Liu [1,2], Yang Luo [1,2,*] and Jinyu Zheng [1,2,*]

[1] Research Institute of Agricultural Resources and Environment, Jilin Academy of Agricultural Science, Changchun 130033, China; hongbingzheng@126.com (H.Z.); ruipinghappy@126.com (R.L.); suipengxiang1990@163.com (P.S.); wanghao19810606@163.com (H.W.); renying690809@126.com (Y.R.); yuan1998e@163.com (Y.Y.); 18243397351@163.com (S.T.); 13294759741@163.com (S.Z.); liuwuren571212@163.com (W.L.)

[2] Key Laboratory of Crop Ecophysiology and Farming System in Northeast China, Ministry of Agriculture, Beijing 100125, China

* Correspondence: nkyly@163.com (Y.L.); 15844052867@163.com (J.Z.)

**Abstract:** A field experiment was conducted to evaluate the effect of different tillage structures on soil physical properties, soil chemical properties, maize root morphological and physiological characteristics, and yield. Four tillage structures were designed. Soil tillage plays a prominent role in agricultural sustainability. The different tillage layer structures affected soil physical properties. An enhancement in the optimal tillage layer structure improved soil structure. The MJ tillage layer structure created an improved soil structure by regulating the soil physical properties so that the soil compaction and soil bulk density would be beneficial for crop growth, increase soil water content, and adjust the soil phrase *R* value and *GSSI*. Soil nutrients are significantly affected by soil depth, with the exception of available potassium. However, soil nutrients are influenced by different tillage layer structures with soil depth. Soil nutrient responses with depth are different for MJ layer treatment compared with other tillage layer structures. Soil organic matter (SOM) is affected with an increase in depth and is significantly influenced by different tillage layer structures, except at 20–30 cm soil depth. MJ treatment increased by 10–20% compared with other tillage layer structures. In addition, QS treatment enhanced the increased pH value in soil profile compared to others. The root morphology characteristics, including root length, root ProjArea, root SurfArea, root AvgDiam, and root volume, were affected by years, depth, and the tillage layer structures. The MJ tillage layer structure enhanced root growth by improving tillage soil structure and increasing soil air and water compared with other tillage layer treatments. Specifically, the MJ layer structure significantly increased root length and root volume via deep tillage. However, the differences in root physiological properties were not significant among treatments. The root dry weight decreased with an increase in soil depth. Most of the roots were mainly distributed in a 0–40 cm soil layer. The MJ treatment enhanced the increase in root dry weight compared with others by breaking the tillage pan layer. Among the different tillage layer structures, the difference in root dry weight was smaller with an increase in soil depth. Moreover, the MJ treatment significantly improved maize yield compared with others. The yield was increased by 14.2% compared to others under MJ treatment via improvements in the soil environment. In addition, the correlation relationship was different among yield and root morphology traits, root physiology traits, soil nutrients, and soil physical traits. So, our results showed that the MJ tillage layer structure is the best tillage structure for increasing maize yield by enhancing soil nutrients, improving the soil environment and root qualities.

**Keywords:** tillage structures; soil physical; chemical properties; root morphology; root physiology; yield

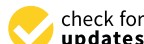



## 1. Introduction

Agriculture is important to societal development and human survival [1]. With worldwide population growth, the sustainable production of food must overcome serious challenges to guarantee the growing global food demand in the future [2]. However, China faces significant challenges in agricultural development, as its population accounts for 22% of the global population, but its arable land accounts for less than 7% of global arable land [3,4]. It is worth noting that the dry land area in Northeast China is large, accounting for approximately 21% of the country's arable land area and over 30% of the country's total grain production [3,4]. However, crop production faces many significant challenges, such as soil degradation, water and nutrient loss, low organic matter content, and fragile physical structures [5,6]. The black soil area in Northeast China is known as the "cornerstone" of maintaining crop yield and national food security in China [7].

It is necessary to take reasonable soil management measures to increase crop yield and protect or maintain soil quality [8]. The soil management system directly intervenes in the production response of crops through changes in soil physicochemical properties and root characteristics [9]. Farming is the process of physically treating soil to improve it with the help of tools [10]. The cultivation system can alter soil moisture content, temperature, aeration, and the degree of mixing of crop residues in the soil matrix, thereby affecting the physical and chemical environment of the soil [11]. This is a key soil management practice that has significant implications for seedbed preparation, root growth stimulation, weed control, soil moisture control, soil temperature control, soil compaction mitigation, soil structure improvement, soil nutrient enhancement, and the incorporation of crop residues and fertilizers [12,13]. In addition, tillage treatment plays an important role in altering soil structure and the distribution of crop residues, thereby affecting the ability of soil microorganisms to degrade soil organic matter and release crop growth nutrients [14]. Therefore, by altering soil characteristics and affecting root growth, it is believed that tillage methods are key factors in the sustainability of planting systems [15].

There are two farming methods available: conservation tillage and conventional tillage [16–18]. Although traditional tillage can loosen the soil surface, promote crop root growth, absorb soil nutrients, and increase crop yield [19], it reduces soil microbial biomass, total carbon, active carbon, total nitrogen, aggregate stability, and sand-free organic matter and increases carbon metabolism [20,21]. Protective tillage practices are divided into no tillage (no tillage), minimal tillage (minimal tillage), cover tillage, ridge tillage, and contour tillage [22]. Conservation tillage has been recognized as one of the most effective soil management measures for the sustainable development of global agriculture [23]. Conservative tillage plays an important role in improving soil structure and maintaining surface soil structure and soil physical conditions [24]. Meanwhile, conservation tillage is recommended as an effective method for maintaining soil moisture in dryland agriculture [25,26].

The purpose of soil cultivation is to prepare soil with sufficient physical conditions for plant growth [27]. Therefore, soil characteristics play an important role in the selection of tillage systems [28,29]. However, farming systems can also affect soil characteristics, including soil structure, soil compaction, soil bulk density, and crust or erosion [30]. Dal et al. [31] pointed out that the negative impact of some farming practices is that they often damage soil structure. Meanwhile, traditional farming reduces the stability of aggregates and increases their bulk density [32,33]. Compared to traditional tillage, the use of conservation tillage may lead to different soil physical properties, as the soil matrix is less disturbed [30]. Logsdon et al. [34] pointed out that when using ridge tillage or switching from traditional practices to no tillage on these or similar deep loess soils, producers do not need to worry about increased compaction. However, the soil bulk density of the corn belt under no tillage treatment was significantly higher, following the pattern of less tillage > no tillage > conventional tillage [30]. The topsoil under NT is usually cooler and wetter and has a higher bulk density (BD), thus exhibiting greater soil strength compared to CT [35]. Fabrizzi et al. [36–38] also reported that no-tillage soils typically have greater resistance and higher packing density than traditional soils to hinder root infiltration.

Soil tillage management has a significant impact on root morphology, root physiology, and root growth and development [39–42]. In addition, root growth may also be indirectly affected by changes in soil properties caused by farming systems [43–46]. By changing soil characteristics and affecting root growth, the cultivation method is a key factor in the sustainability of planting systems [47,48]. The effect of cultivation on the growth of maize roots was previously found in early growth and continued until flowering [49]. Meanwhile, the effects of soil temperature and bulk density changes caused by cultivation on plant growth are mediated by the growth and function of the root system [50]. The aims of this study were to evaluate the effect of different tillage structures on soil physical and chemical properties, determine maize root's morphological and physiological characteristics under different tillage structures, study the yield changes in different tillage layer structures, and classify the relationship yield and soil physical and chemical properties to identify the most beneficial tillage systems.

## 2. Materials and Methods

### 2.1. Site Description

The experiment was conducted during the spring maize growth seasons of 2016 and 2017 at the Gongzhuling Experimental Station of Jilin Academy of Agricultural Sciences in Jilin Province (43°45′ N and 125°01′ E). The local climate is sub-humid, with an average rainfall of 567 mm and an annual average temperature of 6.91 °C. The soil is sandy loam (36.0% sand, 24.5% silt, 39.5% clay). Jilin Province has a continental climate with a wide range of temperatures. The average temperature in Gongzhuling in 2016 was 6.68 °C, and the average temperature in 2017 was 7.05 °C. The annual precipitation values for these years were moderate (with a total precipitation of 890.8 mm in 2016 and 694.3 mm in 2017). The precipitation and temperature data are shown in Figure 1.

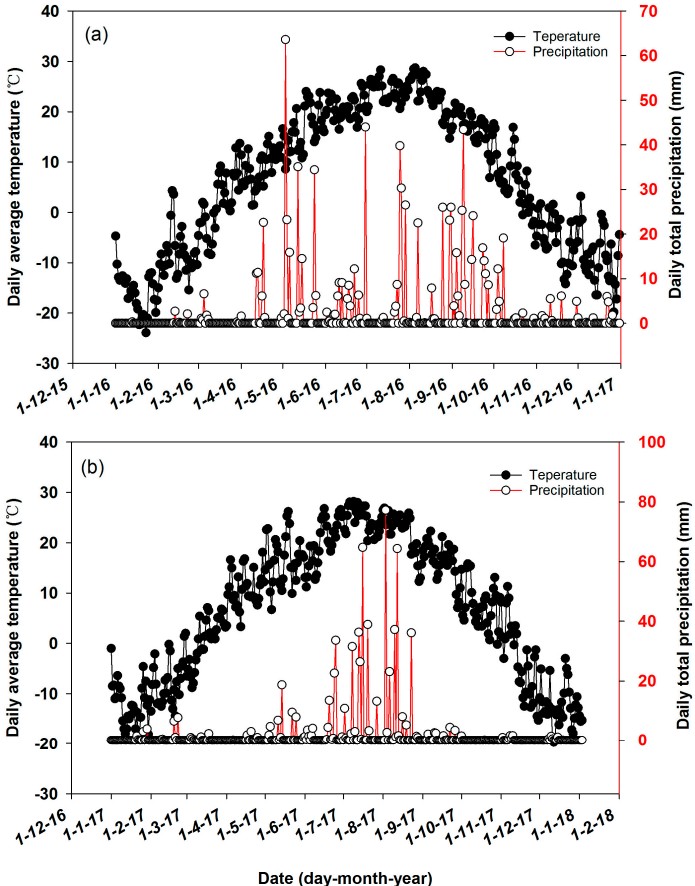

**Figure 1.** Precipitation and temperature at the research site in 2016 (**a**) and 2017 (**b**).

## 2.2. Experimental Design

The experiment was completed in Gongzhuling city, and experimental material is XY998. The experimental zone has a mid-temperate continental monsoon climate with an annual temperature of 4.5 °C and 2800 h cumulative sunshine hours. Meanwhile, the effective accumulated temperature ≥10 °C is 2860 °C d, and the frost-free period is 140 days. The annual precipitation from June to August is 567 mm. The soil type is typical medium black soil with loamy clay. The total nitrogen, total phosphorus, and total potassium contents are 0.15%, 0.05%, and 2.26%, respectively. The available nitrogen, available phosphorus, and available potassium contents are 146.36 mg/kg,13.50 mg/kg, and 152.32 mg/kg. The pH value is 6.5 in 0–20 cm soil depth.

The experiment was conducted during the 2016 and 2017 growing seasons. The experiment was conducted in Gongzhuling city and consisted of four treatments, including compaction seeding soil bed with row soil deep tillage (MJ), softy seeding soil with row soil compaction (MS), compaction seeding soil with row soil compaction (QJ), and softy seeding soil with row soil deep tillage (QS). The treatments were distributed in a completely randomized block design with four treatments and three replications. The experiment device was designed with PVC pipes that were 20 cm in height and 30 cm in diameter. Five PVC pipes that were 20 cm in length were connected by scotch tape and put into the soil according to our experimental requirements. Every treatment involved the use of 64 PVC pipe pillars. The three maize seeds were planted into the soil in a PVC soil pillar in spring. After emerging with 3 leaves, a plant of corn was kept in the PVC soil pillar, and others were cut using a knife. The plant density was 6000 plants per hectare, with 32 cm plant distance and 52 cm row distance. A total of 243 kg/hm$^2$ Controlled release urea was added; 92 kg/hm$^2$ of $P_2O_5$ was added, and 80 kg/hm$^2$ of $K_2O$ was added, and all fertilizers were used as a base fertilizer (one-time application).

Compaction seeding soil bed with row soil deep tillage (MJ): the bulk density was 1.27–1.30 g/cm$^3$, and the soil compaction was 1.00–1.50 Mpa, with 11.5 cm width in the seeding zone; the bulk density was 1.00–1.10 g/cm$^3$, and the soil compaction was 0.10–0.50 Mpa, with 20 cm width in the subsoiling zone. Softy seeding soil with row soil compaction (MS): the bulk density was 1.00–1.10 g/cm$^3$, and the soil compaction was 0.10–0.50 Mpa, with 11.5 cm width in the seeding zone; the bulk density was 1.27–1.30 g/cm$^3$, and the soil compaction was 1.00–1.50 Mpa, with 20 cm width in the compaction row zone. Softy seeding soil with row soil compaction (MS), compaction seeding soil with row soil compaction (QJ): the soil bulk density was 1.27–1.30 g/cm$^3$, and the soil compaction was 1.00–1.50 Mpa, with 53 cm row distance. Softy seeding soil with row soil deep tillage (QS): the soil bulk density was 1.00–1.10 g/cm$^3$, and the soil compaction was 0.10–0.50 Mpa, with 53 cm row distance. The soil water content was calculated for the years 2016 and 2017. The moisture contents of the 0–60 cm soil layers were recorded at 10 cm intervals using a TDR meter.

Deep gouges (4.2 m in length × 2.6 m width × 1 m depth) were made by using a spade in order to put the PVC pipes into the soil. The soil was separated according to the tillage layers. Then, the PVC pipes were put into the deep gouges according to the different treatment types. Overall, 64 PVC soil pillars were used, along with 8 pipes that were arranged horizontally and 8 pipes that were arranged vertically. Finally, soil was returned into the PVC pipes according to the requirements of bulk density and soil compaction. The planting and fertilizer management were same as the field after freeze–thaw in spring. The experimental layout is presented in Figure 2.

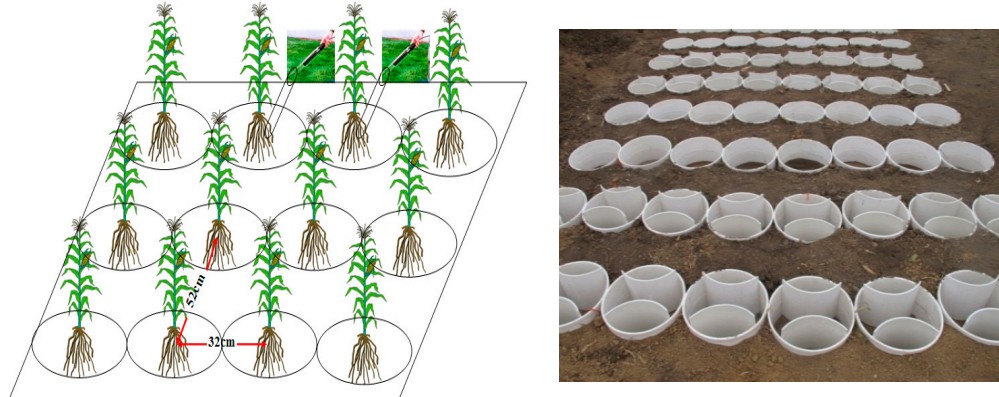

**Figure 2.** The experimental layout.

*2.3. Root Sampling*

The cultivated maize PVC pipes were removed from the soil during the tasseling and silking stage (VT) of maize in 2016 and 2017, respectively. Firstly, the aboveground part of the maize plants was cut off using a knife, and then the PVC pipes were cut off at the joint of each PVC pipe using a knife. According to the height of the PVC pipe, each layer was divided into five layers. Each layer of the PVC pipes in the soil and root were collected together into a wool mesh bag over and over again and placed into a plastic bucket. The mesh bag was washed repeatedly with tap water to remove most of the soil; after rinsing, all of the dirt was cleaned from the mesh bag. Finally, the root of the mesh bag was added into a tray, and some impurities (such as sand) in the root mix were removed so that clean roots could be placed into a plastic bag. The processed names were noted with markers and put into prepared liquid nitrogen tanks to determine the root morphology and physiological indexes.

*2.4. Soil Physical Parameters*

Bulk density (g/cm$^3$) = W/V, W = oven-dry soil weight in grams, V = volume of core in cm$^3$; Total porosity (%) = [1 − (bulk density/particle density)] × 100, particle density = 2.65 g/cm$^3$; $R = \left| \sqrt{0.4 \times (X-50)^2 + (Y-25)^2 + (Z-25)^2} \right|$ X = 100 × (1 − total soil porosity), Y = 100 × soil water content rate, Z = 100 × ( total soil porosity- soil water content rate); $GSSI = [(X_S - 25)X_L X_G]^{0.4769}$, $X_S$, $X_L$, and $X_G$ were the percentage of solid, liquid, and gas phases. Soil compaction was measured by SC-900; Soil water content was measured via the aluminum box weighing method; Soil three-phase was measured using a Soil three-phrase meter (DIK1150).

*2.5. Soil Nutrient Parameters*

The soil samples from 0 to 10 cm, 10 to 20 cm, 20 to 30 cm, 30 to 40 cm, 40 to 50 cm, and 50 to 60 cm depth were collected to separately measure the total nitrogen (TN), total phosphorus (TP), total potassium (TK), available nitrogen (AN), phosphorus (AP), available potassium (AK), organic matter, and pH contents using a soil sampler.

*2.6. Root Morphology and Physiology*

We identified five morphological root traits (Root Length, Root ProjArea, Root SurfArea, Root AvgDiam, and Root Volume). All traits were measured by scanning the root system at 800 dpi using a flatbed scanner. Images were analyzed using WinRhizo Pro 2016 software (2016a, Regent Instruments, Quebec, QC, Canada). The fresh roots were put into a plastic bag and stored in liquid nitrogen tanks to maintain their activity and measure their soluble sugar, soluble protein, POD, and SOD contents in the lab.

*2.7. Maize Grain Yield*

On 2 October 2016 and 2017, corn was harvested manually. The corn ears of each treatment were put into net bags and brought into the laboratory. The ears were put into paper bags and placed in an oven before being dried at 80 °C to a constant weight. The corn yield was determined by manually harvesting each plot over the past two years. Grain and straw samples were air dried on the ground of the threshing field, and the yield was reported at a moisture content of 14%.

*2.8. Data Analysis*

The data were examined via analysis of variance, which was carried using SPSS statistical software (ver. 22.0; SPSS Inc., Chicago, IL, USA). The mean values were compared using the Least Significant Difference (LSD) test.

**3. Results**

*3.1. Effect of Different Tillage Layer Structures on Soil Physical Properties*

The effect of different tillage treatments on soil bulk density was significant at 60 cm soil depth but not at other soil layer depths in all treatments. The soil bulk density of all treatments increased with an increase in soil depth between 0 and 60 cm. The trend of change was significant from 0 to 30 cm, but was not as obvious from 30 cm to 60 cm. The top soil bulk densities of all of the treatments were significantly lower than the other soil layers in terms of soil profile depth. The soil bulk density of the QJ treatment ranged from 1.14 to 1.48 g cm$^{-1}$, and the average mean was 1.38 g cm$^{-1}$ from 0 to 60 cm soil profile depth. The soil bulk density of the MS treatment ranged from 1.08 to 1.52 g cm$^{-1}$, and the average mean was 1.38 g cm$^{-1}$ from 0 to 60 cm soil profile depth. The soil bulk density of the QS treatment ranged from 1.09 to 1.46 g cm$^{-1}$, and the average mean was 1.38 g cm$^{-1}$ from 0 to 60 cm soil profile depth. The soil bulk density of the MJ treatment ranged from 1.16 to 1.51 g cm$^{-1}$, and the average mean was 1.41 g cm$^{-1}$ from 0 to 60 cm soil profile depth (Figure 3).

The effect of different treatments on total soil porosity was significant at 0–10 cm, 10–20 cm, and 50–60 cm. However, that of other soil depths was not significant for all treatments. The change trend of all treatments decreased with the increase in soil depth. The soil porosity of the top 0–10 cm soil depth was significantly greater than the other soil layers. The total soil porosity of the QJ treatment ranged from 43.94 to 53.31%, and the average mean was 46.77% from 0 to 60 cm soil profile depth. The total soil porosity of the MS treatment ranged from 42.40 to 59.05%, and the average mean was 47.66% from 0 to 60 cm soil profile depth. The total soil porosity of the QS treatment ranged from 43.59 to 58.52%, and the average mean was 47.11% from 0 to 60 cm soil profile depth. The total soil porosity of the MJ treatment ranged from 42.91 to 55.96%, and the average mean was 46.12% from 0 to 60 cm soil profile depth (Figure 3).

The profile change of the three-phrase *R* value for all treatments was not obvious with the increase in soil depth. However, the three-phrase *R* values of all treatments were significant in the whole soil profile depth. The three-phrase *R* value of the QJ treatment ranged from 6.09 to 15.43, and the average mean was 10.41 from 0 to 60 cm soil profile depth. The three-phrase *R* value of the MS treatment ranged from 5.77 to 11.62, and the average mean was 10.84 from 0 to 60 cm soil profile depth. The three-phrase *R* value of the QS treatment ranged from 10.42 to 14.70, and the average mean was 12.56 from 0 to 60 cm soil profile depth. The three-phrase *R* value of the MJ treatment ranged from 4.46 to 12.25, and the average mean was 7.95 from 0 to 60 cm soil profile depth (Figure 3).

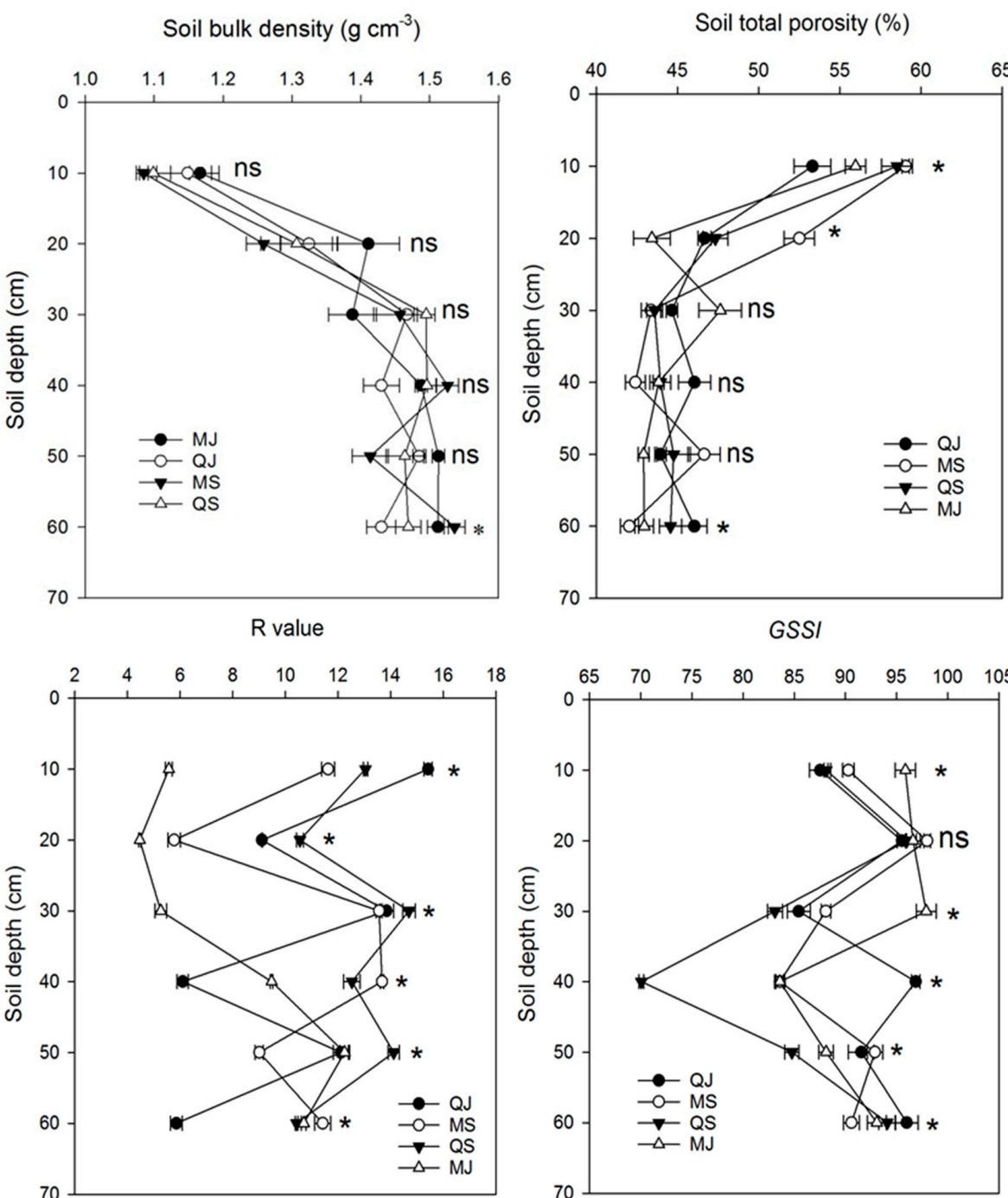

**Figure 3.** Means of soil bulk density, total soil porosity, *R* value, and GSSI at 0–60 cm affected by different tillage structures at the maize harvesting stage (data from 2016 to 2017). "*" was significantly different at *p* < 0.05 by a LSD test and "ns"was not significantly different at *p* < 0.05 by a LSD test.

The profile change of GSSI for all treatments was not obvious with the increase in soil depth. However, the differences in soil profile for all treatments were significant, except for the 10–20 cm soil layer. The GSSI value of the QJ treatment ranged from 85.43 to 95.99, and the average mean was 92.14 from 0 to 60 cm soil profile depth. The GSSI value of the MS treatment ranged from 83.58 to 98.00, and the average mean was 90.57 from 0 to 60 cm soil profile depth. The GSSI value of the QS treatment ranged from 83.11to 95.98, and the average mean was 86.01 from 0 to 60 cm soil profile depth. The GSSI value of the

MJ treatment ranged from 83.58 to 97.90, and the average mean was 92.52 from 0 to 60 cm soil profile depth (Figure 3).

The soil water content profile change of all treatments decreased with increasing depth (Figure 4). The difference in soil water content was significant in the whole profile depth. The soil water content of the QS treatment was significantly higher than that of the other treatments from between 0–20 cm and 50–60 cm, and the percentage increases in soil water content were 0.44–1.59%, 0.81–1.78%, 0.61–1.48%, and 0.96–1.41%, respectively. However, the MJ treatment was significantly greater than other treatments from 30 to 40 cm, and the percentage increase in soil water content was 0.49–0.94%. Nevertheless, the QJ treatment continued to have the lowest water content among all soil profiles.

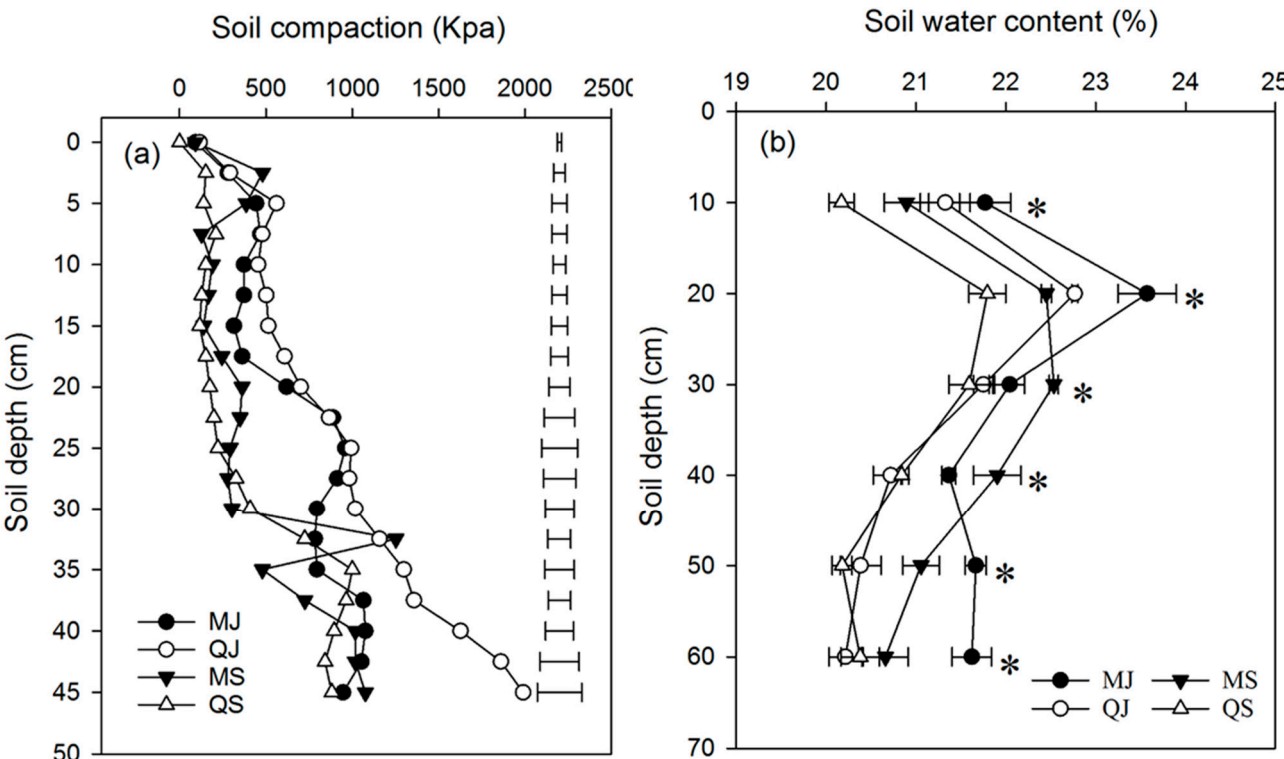

**Figure 4.** The soil water content (**a**) and soil compaction (**b**) profile changes in all treatments at changing depths. Horizontal bars represent ± SE in figure and * was significantly different at $p < 0.05$ by a LSD test.

Soil compaction is an important parameter of soil quality (Figure 4). Out of all of the soil profiles, the soil compaction at 0–45 cm depth increased in all four treatments. The largest increase was in the QJ treatment. Averaged across the depth of measurements, the soil compaction of the QJ treatment (914 kPa) was similar to that of the MS, QS, and MJ treatments, which had average soil compaction values of 472, 404, and 663 kPa, respectively. All treatments have similar tread at 0–35 cm. The QJ treatment continued to have greater soil compaction at 30–45 cm soil depth and increased by 27–109%, 82–170%, and 29–125% compared to MJ, MS, and QS.

### 3.2. Tillage's Effect on Soil Nutrients

The soil nutrient contents of the different treatments decreased with depth in all cases except for the AK treatment. The differences in contents near the surface were higher than at other soil depths, and the lowest soil nutrient concentrations were at the lower soil layers. TN concentrations decreased with the increase in soil depth. It was not significant at 0–30 cm, and it was obvious at 30 cm depth. The differences amongst treatments were significant across all soil profiles. At 0–20 cm soil depth, the TN contents of QS were

significantly higher than the other treatments. However, the lowest TN content for the QS treatment was observed at 20–30 cm and 30–40 cm. At 40–50 cm, QJ and MS were significantly higher QS and MJ, and MJ was lower than the other treatments at 50–60 cm. The value scope of the MS treatment was 0.54 g kg$^{-1}$ to 1.19 g kg$^{-1}$, and the mean value was 0.95 g kg$^{-1}$ in the whole soil profile. The value scope of the QJ treatment was 0.55 g kg$^{-1}$ to 1.23 g kg$^{-1}$, and the mean value was 0.96 g kg$^{-1}$ in the whole soil profile. The value scope of the QS treatment was 0.54 g kg$^{-1}$ to 1.27 g kg$^{-1}$, and the mean value was 0.91 g kg$^{-1}$ in the whole soil profile. The value scope of the MJ treatment was 0.52 g kg$^{-1}$ to 1.23 g kg$^{-1}$, and the mean value was 0.95 g kg$^{-1}$ in the whole soil profile. The results showed four different tillage layer structures in the order of MS > QJ = MJ > QS.

TP concentrations decreased with the increase in soil depth, and significant differences were detected in the soil profiles of the four tillage layer structures. At 0–20 cm soil depth, QS was significantly higher than the other treatments. With the increase in soil depth, the MJ was higher than the other treatments at 20–40 cm. However, the QJ was higher than the other treatments at 40–60 cm soil depth, although this difference was not significant amongst the four treatments. The value scope of the QJ treatment was 0.54 g kg$^{-1}$ to 1.19 g kg$^{-1}$, and the mean value was 0.95 g kg$^{-1}$ in the whole soil profile. The value scope of the MS treatment was 0.55 g kg$^{-1}$ to 1.23 g kg$^{-1}$, and mean value was 0.96 g kg$^{-1}$ in the whole soil profile. The value scope of the QS treatment was 0.54 g kg$^{-1}$ to 1.27 g kg$^{-1}$, and the mean value was 0.91 g kg$^{-1}$ in the whole soil profile. The value scope of the MJ treatment was 0.52 g kg$^{-1}$ to 1.23 g kg$^{-1}$, and the mean value was 0.95 g kg$^{-1}$ in the whole soil profile. The results showed four different tillage layer structures in the order of MS > QJ = MJ > QS (Figure 5).

TK concentrations decreased with the increase in soil depth, and the increasing trend was significant with depth. The differences in the different tillage layer structures were significant with depth across all soil profiles, except at 10–20 cm. At 0–10 cm soil depth, QJ and MS were significantly higher than QS and MJ. At 30–60 cm soil depth, QJ was significantly higher than the other treatments. The value scope of the QJ treatment was 0.34 g kg$^{-1}$ to 0.48 g kg$^{-1}$, and the mean value was 0.41 g kg$^{-1}$ in the whole soil profile. The value scope of the MS treatment was 0.32 g kg$^{-1}$ to 0.50 g kg$^{-1}$, and the mean value was 0.42 g kg$^{-1}$ in the whole soil profile. The value scope of the QS treatment was 0.33 g kg$^{-1}$ to 0.47 g kg$^{-1}$, and the mean value was 0.41 g kg$^{-1}$ in the whole soil profile. The value scope of the MJ treatment was 0.33 g kg$^{-1}$ to 0.46 g kg$^{-1}$, and the mean value was 0.40 g kg$^{-1}$ in the whole soil profile. The results showed that the mean values for the four tillage layer structures were similar(Figure 5).

AN concentrations decreased with the increase in soil depth. The increasing trend was not obvious from 0 cm to 30 cm soil depth but was significant from 40 cm to 60 cm (Figure 5). The difference between the four tillage layer structures was not significant at 0–20 cm. However, the difference was significant at other soil depths. At 20–30 cm soil depth, MS was significantly higher than the other treatments, and QJ was significantly higher than the other three treatments at 30–40 cm and 40–50 cm soil depth. However, MJ was significantly higher than the others at 50–60 cm soil depth. The value scope of the QJ treatment was 43.93 mg kg$^{-1}$ to 116.75 mg kg$^{-1}$, and the mean value was 91.07 mg kg$^{-1}$ in the whole soil profile. The value scope of the MS treatment was 46.01 mg kg$^{-1}$ to 115.53 mg kg$^{-1}$, and the mean value was 89.99 mg kg$^{-1}$ in the whole soil profile. The value scope of the QS treatment was 45.28 mg kg$^{-1}$ to 116.01 mg kg$^{-1}$, and the mean value was 82.95 mg kg$^{-1}$ in the whole soil profile. The results showed four different tillage layer structures in the order of QJ > MS > MS > MJ.

AP concentrations decreased with the increase in soil depth. At 0–10 cm, QS was significantly higher than the others. However, QJ was significantly higher than the others at 30–40 cm, 40–50 cm, and 50–60 cm. The value scope of the QJ treatment was 5.88 mg kg$^{-1}$ to 16.30 mg kg$^{-1}$, and the mean value was 11.56 mg kg$^{-1}$ in the whole soil profile (Figure 5). The value scope of the MS treatment was 6.17 mg kg$^{-1}$ to 16.65 mg kg$^{-1}$, and the mean value was 11.16 mg kg$^{-1}$ in the whole soil profile. The value scope of the QS treatment was

4.96 mg kg$^{-1}$ to 17.43 mg kg$^{-1}$, and the mean value was 11.07 mg kg$^{-1}$ in the whole soil profile. The value scope of the MJ treatment was 5.60 mg kg$^{-1}$ to 16.08 mg kg$^{-1}$, and the mean value was 10.91 g kg$^{-1}$ in the whole soil profile. The results showed four different tillage layer structures in the order of QJ = MS = MS > MJ.

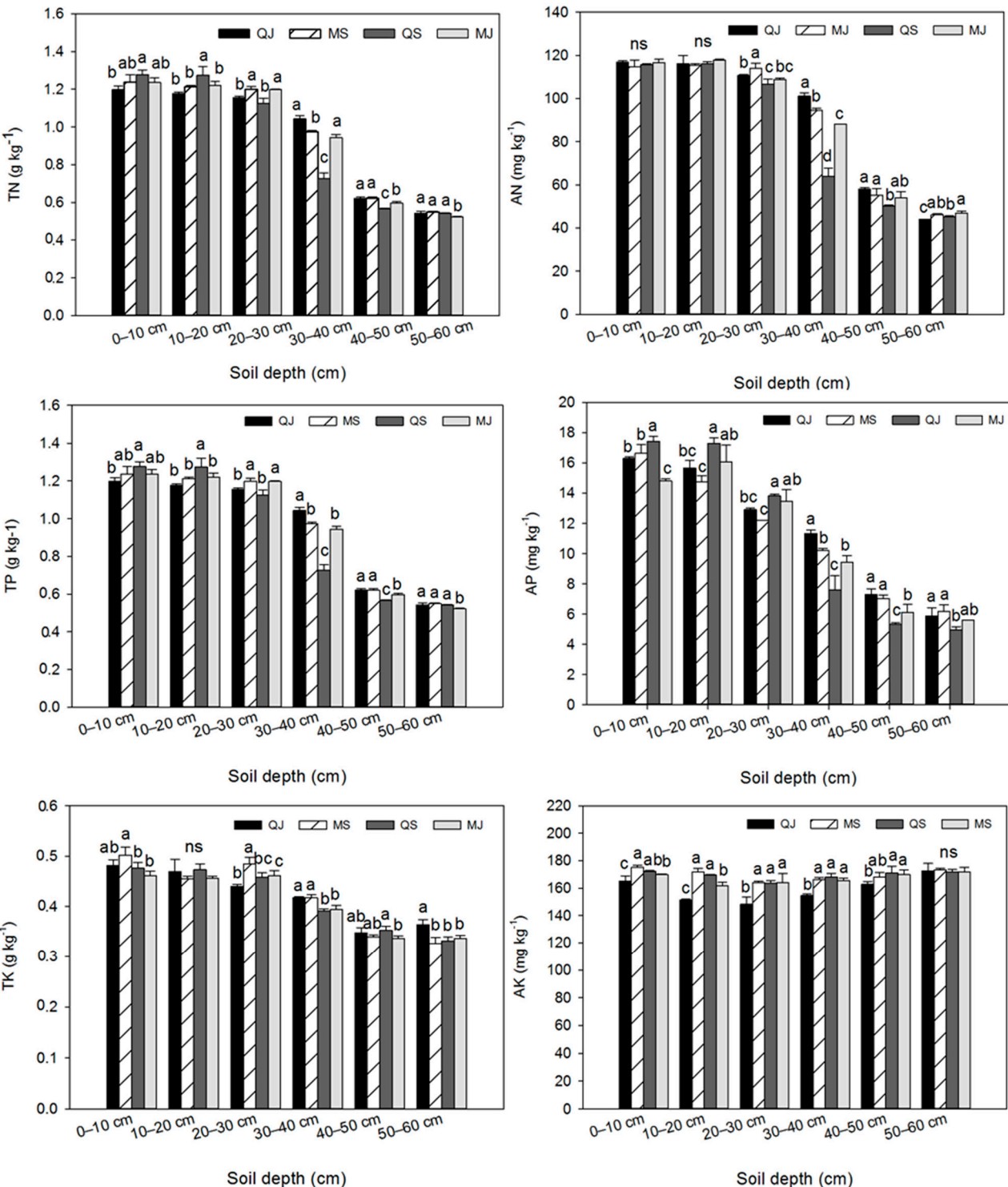

**Figure 5.** Tillage's effect on soil nutrient content in soil profile. Vertical bars represent ± SE and Different lowercase letters indicate significant difference among different treatments at 0.05 levels.

AK concentration was not significant with the increase in soil depth. However, the differences in the different tillage layer structures were significant in the whole soil profile except at 50–60 cm soil depth. MS was significantly higher than the other treatments at 0–10 cm, 10–20 cm, and 20–30 cm. QS was significantly higher than the other treatments at 30–40 cm and 40–50 cm. The value scope of the QJ treatment was 148.25 mg kg$^{-1}$ to 172.58 mg kg$^{-1}$, and the mean value was 159.13 mg kg$^{-1}$ in the whole soil profile. The value scope of the MS treatment was 164.04 mg kg$^{-1}$ to 175.11 mg kg$^{-1}$, and the mean value was 169.75 mg kg$^{-1}$ in the whole soil profile. The value scope of the QS treatment was 163.73 mg kg$^{-1}$ to 172.23 mg kg$^{-1}$, and the mean value was 169.39 mg kg$^{-1}$ in the whole soil profile. The value scope of the MJ treatment was 161.86 mg kg$^{-1}$ to 171.62 mg kg$^{-1}$, and the mean value was 167.15 g kg$^{-1}$ in the whole soil profile. The results showed four different tillage layer structures in the order of MS = QS> MJ> QJ (Figure 5).

The significant effects exerted by the different tillage layer structures are displayed in Figure 6. The change trend was not significant from 0 to 30 cm soil depth, and the SOM decreased significantly at 30–60 cm depth. At 10 cm soil depth, QS had a significantly higher soil organic matter concentration value than the other treatments, and the increase in SOM ranged between 3.81 and 7.20%. At 20 cm soil depth, QS had a significantly higher soil organic matter concentration value than the other treatments, and the increase in SOM ranged between 2.47 and 6.09%. At 40 cm soil depth, QJ had a significantly higher soil organic matter concentration value than the other treatments, and the increase in SOM ranged between 8.96 and 45.55%. At 50 cm soil depth, QJ had a significantly higher soil organic matter concentration value than the other treatments, and the increase in SOM ranged between 6.18 and 13.80%. At 60 cm soil depth, QS had a significantly higher soil organic matter concentration value than the other treatments, and the increase in SOM ranged between 0.78 and 8.98%.

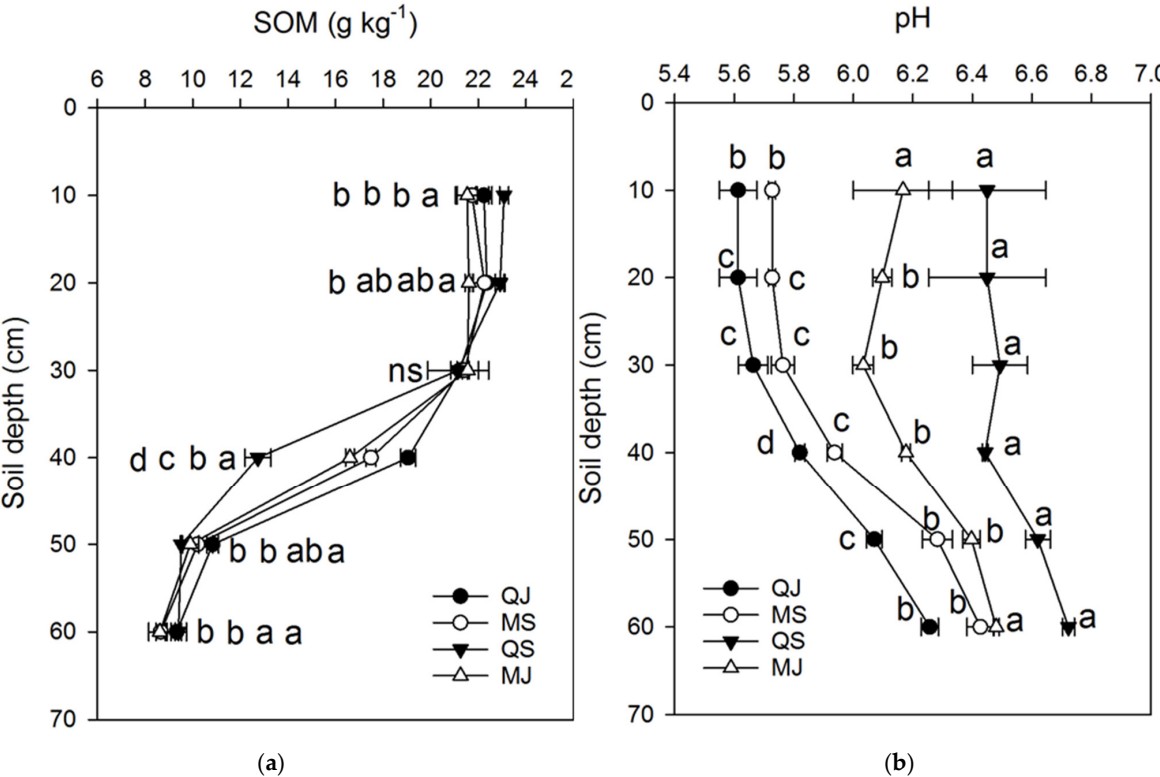

**Figure 6.** Tillage's effect on soil organic matter (**a**) and pH (**b**) content in soil profile. Horizontal bars represent ± SE and different lowercase letters indicate significant difference among different treatments at 0.05 levels.

Soil pH value was affected by the different tillage layer structures, and the trend of soil pH value is shown in Figure 6. The QS had a significantly higher soil pH value than the other treatments in whole soil profile. The increase in soil pH value was not obvious from 0 to 30 cm soil depth. However, the increase in soil pH value was observed to be significant at 30–60 cm soil depth. Across all soil profiles, QS had a significantly higher soil pH value than the other treatments, and the increase in soil pH value ranged between 3.75 and 14.90%.

### 3.3. Effect of Different Tillage Layer Structures on Root Morphology

From Table 1, it can be seen that root length, root ProjArea, root SurfArea, root AvgDiam, and root volume are all significantly affected by the year, depth, and treatments at the VT and R1 stage, respectively. In the VT stage, the root length in 2016 was significantly higher than that in 2017. The root length decreased with depth, and there were significant differences amongst the different depths. The root length of QJ and QS were significantly higher than that of MJ and MS. The root length was significantly affected by year (Y), depth (D), and treatments (T) Y×D, Y×T, D×T, and Y×D×T at 0.05 level. The root ProjArea in 2016 was significantly higher than that in 2017. The root ProjArea at 0–20 cm soil depth was significantly higher than that at other depths. The root ProjArea values of QJ, MJ, and MS were significantly higher than that of QS. The root ProjArea was significantly affected by year (Y), depth (D), and treatments (T) Y×D and Y×T (except D×T and Y×D×T) at 0.05 level. The root SurfArea in 2016 was significantly higher than that in 2017. The root SurfArea at 0–20 cm soil depth was significantly higher than that at other depths. The root SurfArea values of QJ and MJ were significantly than that of QS and MS. The root SurfArea was significantly affected by year (Y), depth (D), and treatments (T) Y×D and Y×T D×T (except Y×D×T) at 0.05 level. The root AvgDiam in 2017 was significantly higher than that in 2016. The root AvgDiam at 0–20 cm, 60–80 cm, and 80–100 cm were significantly higher than at 20–40 cm and 40–60 cm soil depths. The root AvgDiam values of MJ and MS were significantly higher than that of QJ and QS. The root AvgDiam was significantly affected by year (Y), depth (D), and treatments (T) Y×D, Y×T, D×T, and Y×D×T at 0.05 level. The root Volume in 2017 was significantly higher than that in 2016. The root Volume at 80–100 cm was significantly higher than that at other soil depths. The root Volume of MJ was significantly higher than that of the other treatments. The root Volume was significantly affected by year (Y), depth (D), and treatments (T) Y×D, Y×T, D×T, and Y×D×T at 0.05 level.

In the R1 stage, the root length in 2016 was significantly higher than that in 2017. The root length at 0–20 cm soil depth was significantly higher than that at 20–40 cm, 40–60 cm, 60–80 cm, and 80–100 cm soil depth. The root length of MS was significantly higher than that of the other treatments. The root length was significantly affected by year (Y), depth (D), and treatment (T) (except Y×D, Y×T, D×T, and Y×D×T) at 0.05 level. The root ProjArea in 2016 was considerably higher than that in 2017. The root ProjArea at 0–20 soil depth was significantly higher than that at other depths. The root ProjArea of MS was significantly higher than that of the other treatments. The root ProjArea was significantly affected by year (Y), depth (D), treatment (T), and Y×D, except Y×T, D×T, and Y×D×T at 0.05 level. The root SurfArea in 2016 was significantly higher than that in 2017. The root SurfArea at 0–20 cm soil depth was significantly higher than that at 20–40 cm, 40–60 cm, 60–80 cm, and 80–100 cm soil depth. The root SurfArea of MS was significantly higher than that of the other treatments. The root SurfArea was significantly affected by year (Y), depth (D), and treatment (T) Y×D (but not Y×T, D×T, and Y×D×T) at 0.05 level. The root AvgDiam in 2017 was significantly higher than that in 2016. The root AvgDiam at 0–20 cm and 80–100 cm was significantly higher than that at 20–40 cm, 40–60 cm, and 60–80 cm soil depth. There were no significant differences amongst the different treatments. The root SurfArea was significantly affected by year (Y), depth (D), and Y×D (except treatments (T), Y×T, D×T, and Y×D×T) at 0.05 level. The root Volume in 2016 was significantly higher than that in 2017. The root Volume at 0–20 cm soil depth was significantly higher

than that at 20–40 cm, 40–60 cm, 60–80 cm, and 80–100 cm soil depth. The root Volume values of QS and MJ were significantly higher than that of QJ and MS. The root Volume was significantly affected by year (Y), depth (D), and treatments (T) Y×D, Y×T, D×T, and Y×D×T at 0.05 level.

**Table 1.** Effect of tillage systems on maize roots.

| | | VT | | | | | R1 | | | | |
|---|---|---|---|---|---|---|---|---|---|---|---|
| | | Root Length (cm) | Root Proj. Area (cm$^2$) | Root Surf. Area (cm$^2$) | Root Avg. Diam. (mm) | Root Volume (cm$^3$) | Root Length (cm) | Root Proj. Area (cm$^2$) | Root Surf. Area (cm$^2$) | Root Avg. Diam (mm) | Root Volume (cm$^3$) |
| Year (Y) | 2016 | 1286.20 a | 47.00 a | 34.17 a | 0.76 b | 12.14 b | 1385.8 a | 47.51 a | 35.02 a | 0.75 b | 44.76 a |
| | 2017 | 1051.14 b | 38.09 b | 27.59 b | 1.44 a | 62.91 a | 1059.5 b | 38.30 b | 27.63 b | 1.35 a | 18.40 b |
| Depth (D) | 0–20 | 1512.00 a | 58.47 a | 42.51 a | 1.27 a | 42.72 b | 1693.5 a | 58.90 a | 43.12 a | 1.19 ab | 43.11 a |
| | 20–40 | 1105.04 b | 38.91 b | 28.34 b | 0.72 b | 23.63 c | 1118.7 b | 39.06 b | 28.58 b | 0.71 d | 19.90 e |
| | 40–60 | 1089.93 c | 38.57 c | 28.02 c | 0.82 b | 18.25 d | 1111.0 b | 39.02 b | 28.51 b | 0.97 c | 29.89 c |
| | 60–80 | 1069.63 d | 38.39 cd | 27.83 cd | 1.42 a | 41.64 b | 1100.0 c | 38.87 bc | 28.29 c | 1.03 bc | 37.14 b |
| | 80–100 | 1066.75 d | 38.36 d | 27.73 d | 1.27 a | 61.38 a | 1089.6 d | 38.69 c | 28.12 d | 1.36 a | 27.87 d |
| Treatment (T) | QJ | 1209.67 a | 42.65 a | 31.24 a | 0.97 b | 38.82 b | 1219.2 b | 42.88 b | 31.31 b | 1.05 a | 30.87 b |
| | MJ | 1195.24 b | 42.61 a | 31.04 b | 1.30 a | 40.63 a | 1218.9 b | 42.82 b | 31.29 b | 1.09 a | 31.98 a |
| | MS | 1072.87 c | 42.42 ab | 30.59 c | 1.15 ab | 32.38 b | 1232.8 a | 43.08 a | 31.46 a | 1.03 a | 30.85 b |
| | QS | 1196.90 b | 42.49 b | 30.66 c | 0.99 b | 38.26 c | 1219.0 b | 42.85 b | 31.24 b | 1.03 a | 32.62 a |
| analysis of variance | Y | * | * | * | * | * | * | * | * | * | * |
| | D | * | * | * | * | * | * | * | * | * | * |
| | T | * | * | * | * | * | * | * | * | ns | * |
| | Y×D | * | * | * | * | * | * | * | * | * | * |
| | Y×T | * | * | * | * | * | ns | * | ns | ns | * |
| | D×T | * | ns | * | * | * | ns | ns | ns | ns | * |
| | Y×D×T | * | ns | ns | * | * | ns | ns | ns | ns | * |

Numbers followed by the different letter were significantly different at *p* < 0.05 by a LSD test. "*" significance at the 0.05 level of probability. "ns" was not significantly different at *p* < 0.05 by a LSD test.

### 3.4. Tillage's Effect on Root Physiological Properties

The root physiological traits were almost affected by the stages. However, soluble sugar, soluble protein, POD, and SOD were significantly affected by the different tillage layer structures (Table 2). The soluble sugar content was significantly higher in QJ than in the other treatments and only affected by the type of treatment (not other factors). The soluble protein content was significantly higher in MS than in the other treatments and only affected by the type of treatment (not other factors). The POD content was significantly higher in MS than in the other treatments and only affected by the type of treatment (not other factors). The SOD content was significantly affected by Y×S, S×T, and Y×S×T.

**Table 2.** Effect of tillage on corn root physiological traits.

| | | Soluble Sugar % | Soluble Protein mg·g$^{-1}$ | POD u·g$^{-1}$ | SOD u·g$^{-1}$ |
|---|---|---|---|---|---|
| Year (Y) | 2016 | 0.003 a | 7.323 a | 330.14 a | 223.62 a |
| | 2017 | 0.003 a | 8.254 a | 313.50 a | 270.42 a |
| Stage (S) | VT | 0.003 a | 7.914 a | 327.52 a | 241.16 a |
| | R1 | 0.003 a | 7.663 a | 316.12 a | 252.88 a |
| Treatment (T) | QJ | 0.005 a | 7.634 bc | 309.13 b | 251.74 a |
| | MJ | 0.002 bc | 8.503 bc | 267.04 b | 235.86 a |
| | MS | 0.003 bc | 11.204 a | 411.24 a | 260.27 a |
| | QS | 0.001 c | 3.815 c | 299.86 b | 240.22 a |
| analysis of variance | Y | ns | ns | ns | ns |
| | S | ns | ns | ns | ns |
| | T | * | * | * | ns |
| | Y×S | ns | ns | ns | * |
| | Y×T | ns | ns | ns | ns |
| | S×T | ns | ns | ns | * |
| | Y×S×T | ns | ns | ns | * |

Numbers followed by the different letter were significantly different at $p < 0.05$ by a LSD test." *" Significance at the 0.05 level of probability. "ns"was not significantly different at $p < 0.05$ by a LSD test.

### 3.5. Tillage's Effect on Root Dry Weight and Yield

The root dry weight of different tillage layer structures decreased with the increase in soil depth in 2016 and 2017, respectively (Figure 7). The proportions of root dry weight for all treatments were 78.27% and 61.70% in 2016 and 2017, respectively. However, the differences in root dry weight were not significant among the different treatments at 10–20 cm, 20–30 cm, and 30–40 cm, respectively. The different tillage layer structures were significantly different at 50–60 cm. In 2017, the different tillage layer structures were not significant at 0–10 cm, 60–80 cm, and 80–100 cm. However, the differences were significant at 20–40 cm and 40–60 cm. At 20–40 cm soil depth, MJ and QS were significantly higher than QJ and MS. At 40–60 cm soil depth, MJ was significantly higher than other treatments. The different tillage layer structures had significant effects on grain yield across both years. The yield of the MJ treatment was significantly higher than the others in 2016 and 2017, respectively. In 2016, the yield of MJ increased by 9.04%, 23.80%, and 26.06% compared to QJ, MS, and QS, respectively (Figure 8). However, the yield of MJ increased by 11.50%, 10.10%, and 38.01% compared to QJ, MS, and QS, respectively (Figure 9).

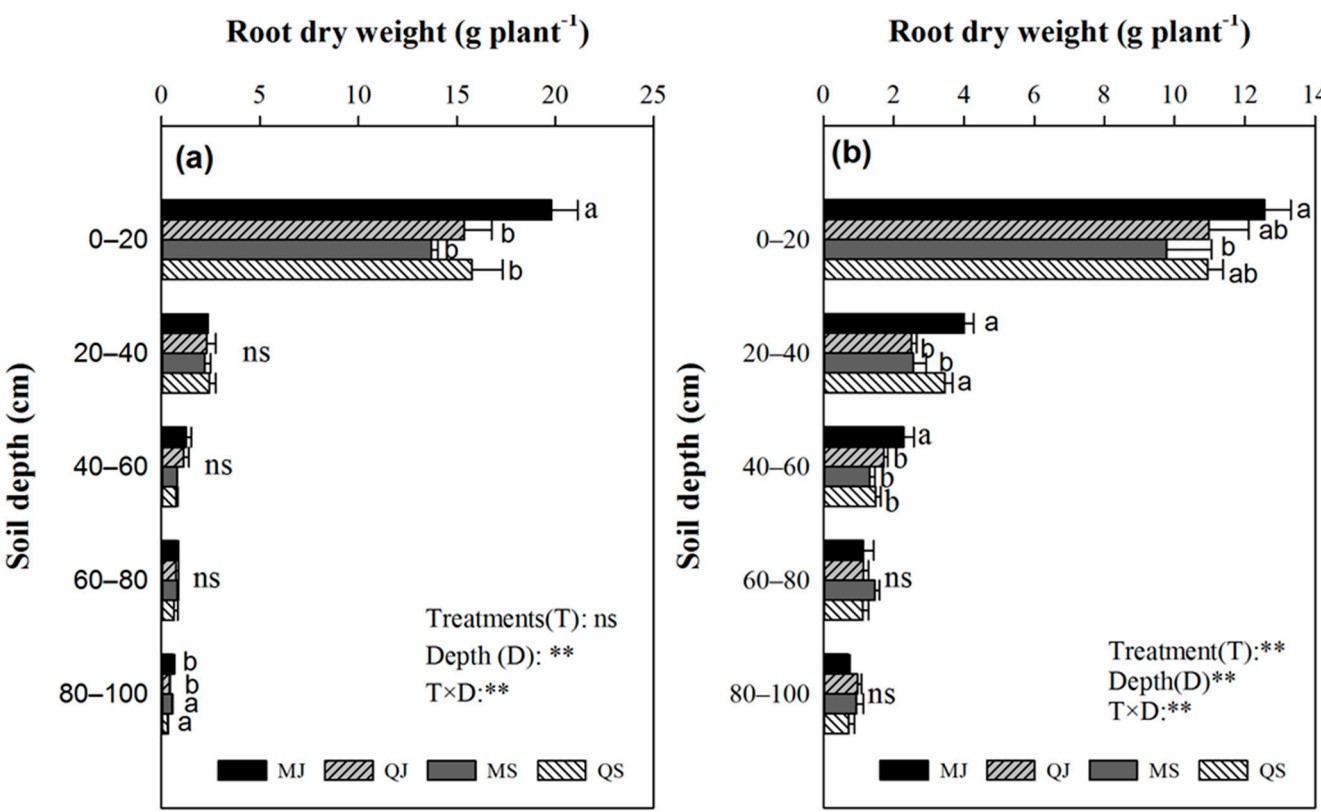

**Figure 7.** Tillage's effect on root dry weight (g per plant) in 2016 (**a**) and 2017 (**b**). Different lowercase letters indicate significant difference among different treatments at 0.05 levels. Horizontal bars represent ± SE. "*" significance at the 0.01 level of probability.

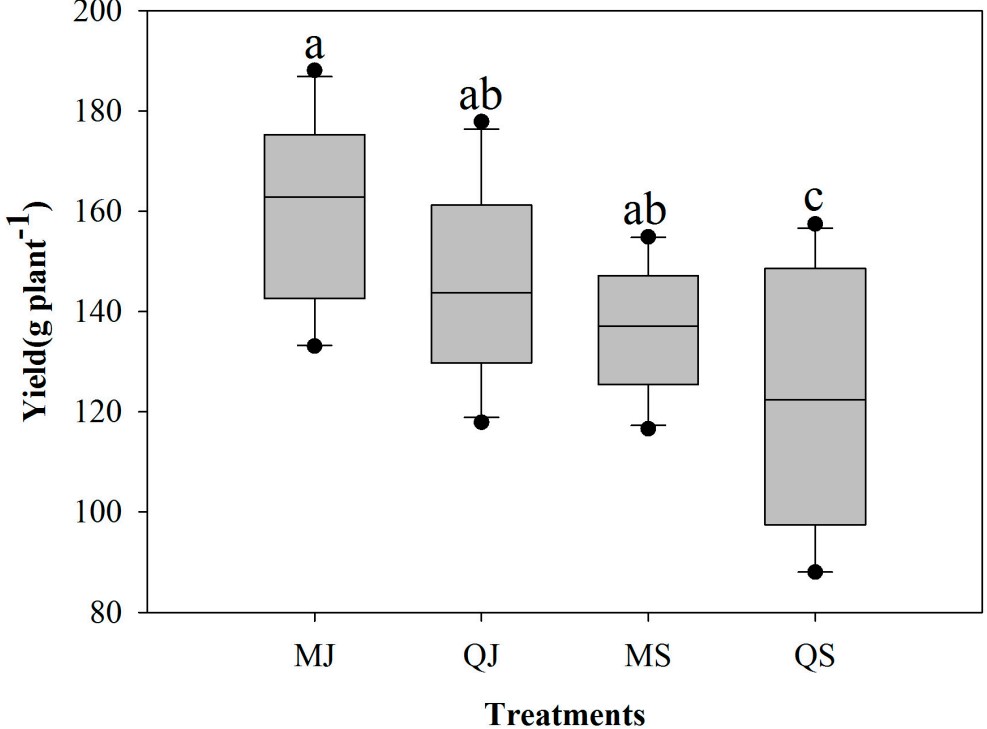

**Figure 8.** Effects of tillage structures on grain yield (2016–2017). Different lowercase letters on the vertical bars indicate significant difference among different treatments at 0.05 levels.

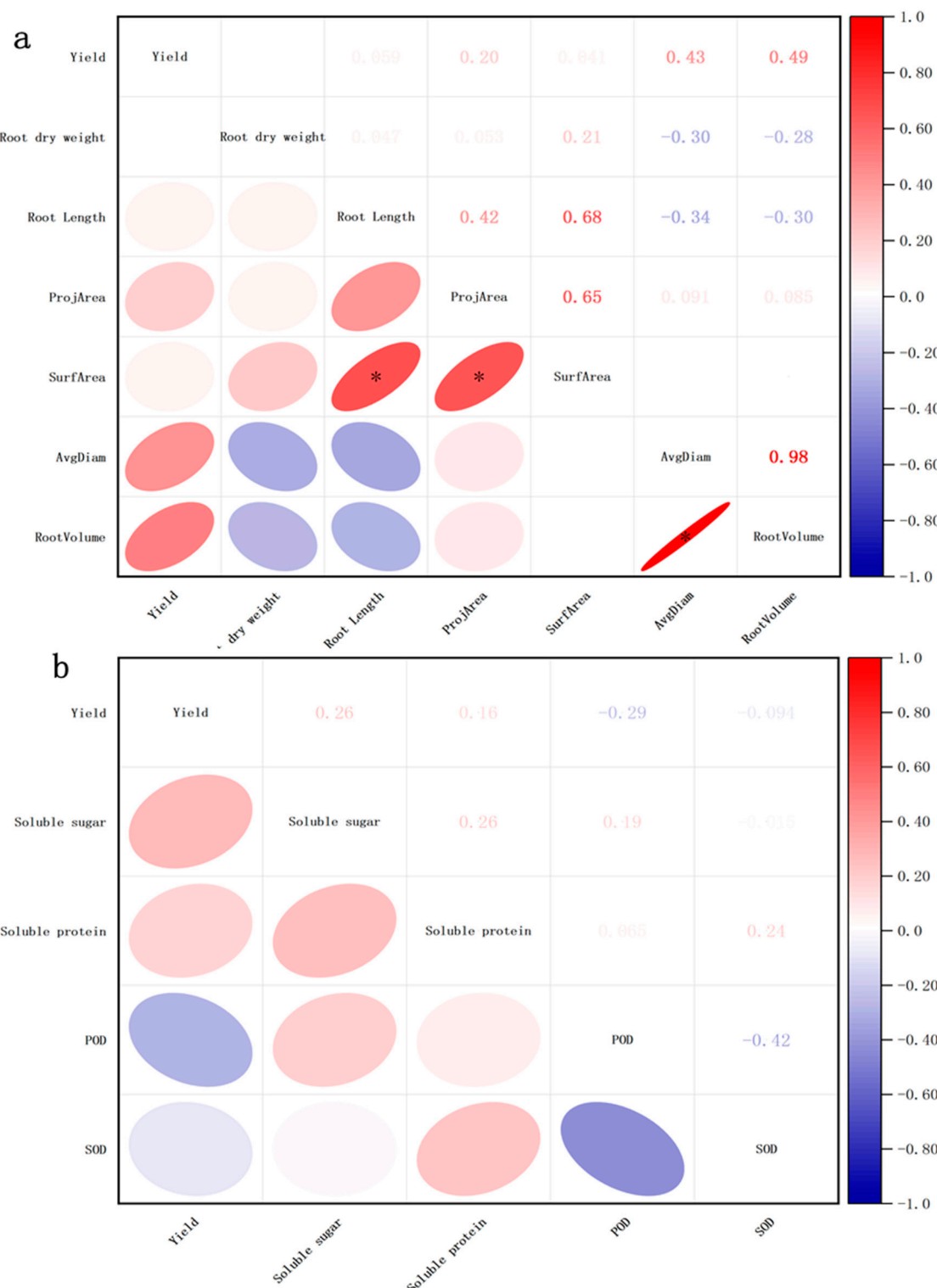

**Figure 9.** *Cont.*

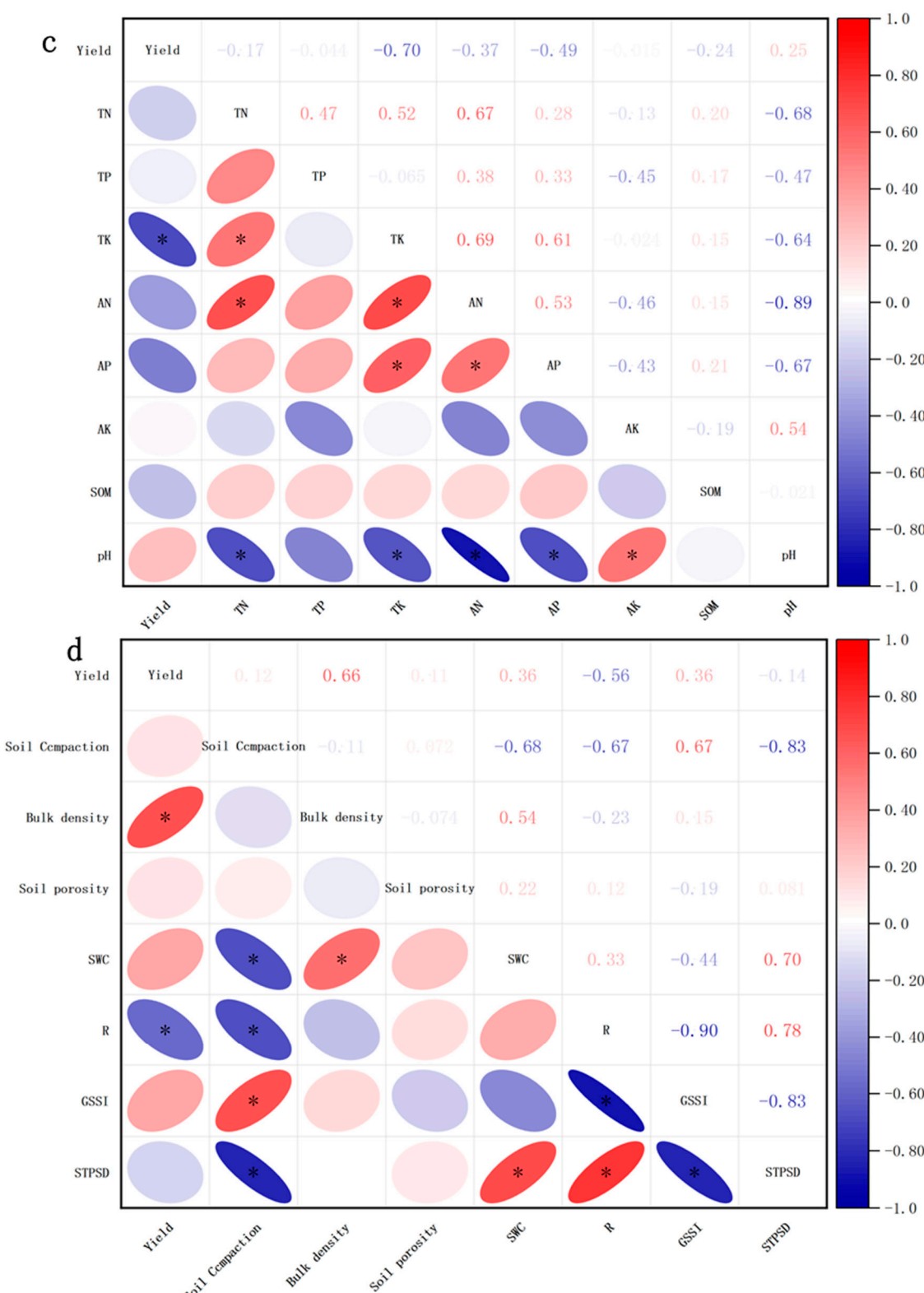

**Figure 9.** Relationship analysis of yield and root morphology (**a**) traits, root physiology traits (**b**), soil nutrients (**c**), and soil physical traits (**d**). "*" Significance at the 0.05 level of probability.

### 3.6. Correlation Analysis of Yield and Other Parameters

There was no significant correlation between the yield and morphology parameters; however, there was a significant positive correlation between the SurfArea and Root

length and ProjArea (Figure 9a). Yield was not significant with physiological parameters, including soluble sugar, soluble protein, POD, and SOD (Figure 9b). There was a significant negative correlation between the yield and TK, and yield was not correlated with other parameters, and there were different correlations among other parameters (Figure 9c). There was a positive correlation between yield and soil bulk density, and there was a negative correlation between yield and R; however, there were different correlations among the different parameters (Figure 9d).

## 4. Discussion

### 4.1. Soil Physical Properties

Improving soil physical properties is important for soil conservation and crop yield enhancement [51]. Soil physical properties are especially positively influenced by different crop rotations, cover cropping, conversation tillage systems, and chemical and organic fertilizers [52–54]. Rotation can reduce stacking density, increase soil aggregate size, and improve water retention by increasing crop residues in the soil and crop residues [10,55,56]. After tillage and harvesting operations, the soil permeability resistance, dry bulk density, and moisture content at all depths are significantly affected by tillage [57]. Aikins et al. [58] showed that after tillage and harvesting operations at a soil depth of 0–60 cm, the zero tillage system produced the highest soil permeability resistance. In addition, the bulk density (Db) of soil changes significantly with the application of different combinations of chemical and organic fertilizers [59]. Our results also proved that soil physical properties can be affected by many factors, such as different tillage layer structures. The MJ layer structure is a better tillage structure that can decrease soil bulk density and soil compaction and increase soil porosity by increasing deep tillage depth. Khurshid et al. [60] showed that Db is an inferior organic fertilizer than inorganic fertilizer. With the continuous application of inorganic and organic fertilizers, the soil particle density (Dp) of surface soil samples remains basically unchanged [61]. Meanwhile, this tillage layer can improve soil water content by enhancing the rainfall infiltration to manipulate the soil structure by improving the three-phrase soil. Our results are consistent with those published by experts in this regard. However, QJ did not significantly increase soil compaction and bulk density in our study compared to the studies of others [62,63].

### 4.2. Soil Nutrient Characteristics

Rotation is the cheapest and most effective method to increase crop yield and soil fertility [64]. The soil nutrient characteristics typically affected by tillage systems include pH, CEC, exchangeable cations, and total soil nitrogen [65]. Conservation tillage, especially MT, is superior to CT in soil chemical improvement [66–68]. Covering crops can protect soil from erosion, reduce N and P losses, increase soil C, reduce runoff, inhibit pests, and support animals that benefit from soil [69,70]. According to reports, the rotation of legumes and covering crops can affect soil nutrient status [71]. Due to differences in crop residues and soil organic matter mineralization rates, crop rotation and nitrogen fertilizer can affect SOC sequestration in cultivated and non-cultivated soils [72]. However, our results show that soil nutrients are significantly affected by soil depth (with the exception of available potassium). However, soil nutrients are influenced by different tillage layer structures with soil depth. Soil nutrient responses with depth are different for the MJ layer treatment compared with other tillage layer structures. Soil organic matter (SOM) values are affected with increasing depth and significantly influenced by different tillage layer structures (except at 20–30 cm soil depth). The MJ treatment increases SOM by 10–20% compared with other tillage layer structures. In addition, in our study, QS treatment more effectively enhanced the increase in the pH value of the soil profile compared to the other treatments. Covering crops is usually incorporated into the planting system as a nutritional management tool [73]. The benefits of legume-covered crops in crop rotation have long been recognized and are mainly attributed to the contribution of nitrogen to subsequent crops [74].

*4.3. Root Morphological and Physiological Traits*

The morphology of maize roots in the early stages of growth is influenced by tillage intensity [75]. The root system is an important component of plants, regulating many aspects of aboveground growth and development. Appropriate crop management can significantly improve the ultrastructure of root tip cells and increase root length density, and thereby increasing grain filling rate, yield, and water use efficiency [52,76]. The poor growth of roots and buds in maize seedlings may be due to the lower surface temperature of NT rather than mechanical impedance [77,78]. The increase in topsoil stacking density during NT treatment may only limit root growth to a limited extent and is more pronounced in fine-grained soil [79]. Our results show that root morphology characteristics such as root length, root ProjArea, root SurfArea, root AvgDiam, and root volume are affected according to the year, depth, and tillage layer structure. The MJ layer structure can enhance root growth by improving tillage soil structure and increasing soil air and water more effectively than other tillage layer treatments. Specifically, the MJ layer structure increased root length and root volume significantly in deep soil. However, the difference in the root physiological properties was not significant among treatments. The effect of cultivation on the growth of maize roots was previously found in early growth and persisted until flowering in our experiment [80].

The average root dry matter (RDM) of corn in the entire soil profile and growth period is affected by tillage, and there are significant differences in RDM for each soil layer under different tillage treatments [81]. Nitrogen fertilizer significantly reduced the root/shoot weight ratio, but tillage did not significantly change the root/shoot weight ratio [82]. DeFelice et al. [83] reported that tillage to 50 cm in subsoil significantly increased the dry weight of spring maize roots at soil depths of 0–80 cm, especially in deep soil [60]. Our results showed that the root dry weight decreases with increasing soil depth. Most of our roots were mainly distributed at 0–40 cm soil depth. The MJ treatment enhanced the increase in root dry weight by breaking the tillage pan layer more effectively than the others. The difference in root dry weight would have been smaller with increasing soil depth among the different tillage layer structures. In the southern and western regions, the yield of no-tillage is often higher than that of traditional tillage [63]. When maize is rotated, minimum tillage can produce the same grain yield as traditional tillage [84,85]. The MJ treatment improves maize yield significantly compared to other treatments. The yield is increased by 14.2% compared to others under The MJ treatment via improving the soil environment and soil function. So, our results show that the MJ tillage layer structure is the best tillage structure for increasing maize yield by enhancing soil nutrients, improving soil environment and root qualities. In addition, the correlation relationships were different among yield and root morphology traits, root physiology traits, soil nutrients, and soil physical traits (Figure 9).

## 5. Conclusions

Soil tillage plays a prominent role in agricultural sustainability. Different tillage layer structures affect soil physical properties. An enhancement in the optimal tillage layer structure improved soil structure. The MJ tillage layer structure could create better soil structures by regulating the soil physical properties, which would be beneficial for crop growth, increase soil water content, and adjust the soil phrase *R* value and *GSSI*. Soil nutrients are significantly affected by soil depth (except available potassium). However, soil nutrients are influenced by different tillage layer structures with soil depth. Soil nutrient responses with depth are different for MJ tillage layer treatment compared with other tillage layer structures. Soil organic matter (SOM) values are affected with increasing depth and significantly influenced by different tillage layer structures (except at 20–30 cm soil depth). The MJ tillage treatment increases SOM by 10–20% compared with other tillage layer structures. In addition, QS treatment enhanced the increase in pH value in the soil profile more effectively than the other treatments. Root morphology characteristics such as root length, root ProjArea, root SurfArea, root AvgDiam, and root volume are affected according

to the year, depth, and tillage layer structure. The MJ layer structure enhanced root growth by improving tillage soil structure and increasing soil air and water compared with other tillage layer treatments. Specifically, the MJ tillage layer structure significantly increased root length and root volume in deep soil. However, the difference in root physiological properties was not significant among the different treatments. Root dry weight decreases with increasing soil depth. Most of the roots were mainly distributed at 0–40 cm soil depth. MJ tillage treatment enhanced the increase in root dry weight by breaking the tillage pan layer more effectively than the others. The difference in root dry weight became smaller with increasing soil depth among the different tillage layer structures. Moreover, MJ tillage treatment significantly improved maize yield compared with the other treatments. The yield was increased by 14.2% compared to the other treatments under the MJ tillage treatment via improvements in the soil environment and soil function. So, our results show that the MJ tillage layer structure is the best tillage structure for increasing maize yield by enhancing soil nutrients, improving the soil environment and root qualities.

**Author Contributions:** Conceptualization, H.W. and W.L.; Methodology, S.Z. and Y.L.; Software, H.Z., R.L., P.S. and J.Z.; Validation, S.T.; Formal analysis, H.Z.; Investigation, S.T., W.L. and J.Z.; Resources, P.S., H.W. and Y.R.; Data curation, H.Z. and Y.R.; Writing—original draft, Y.L.; Writing—review and editing, R.L. and Y.Y.; Visualization, Y.Y.; Funding acquisition, S.Z. and Y.L. All authors have read and agreed to the published version of the manuscript.

**Funding:** This study was funded by the Chinese Academy of Sciences Strategic Pilot Technology Project (XDA28080204), the National Natural Science Foundation of China (31501248), and the National Key Technology Research and Development Program of the Ministry of Science and Technology of China (2016YFD03002).

**Data Availability Statement:** The data presented in this study are available on request from the first author.

**Conflicts of Interest:** The authors declare no conflict of interest.

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
