# Peer review of "Potential Mechanism of Optimal Tillage Layer Structure for Improving Maize Yield and Enhancing Root Growth in Northeast China"

_land, doi:10.3390/land12091798_

Round 1

Reviewer 1 Report

There is a lot of interesting data in the manuscript on Potential mechanism of optimal tillage layer structure for improving maize yield and enhancing root growth in Northeast China. While the experimental design is balanced/appropriate and detailed time courses of various antioxidant enzyme activities are presented  at different critical stages of development over 2 years) in addition to  root growth  and agronomic data (yield components). However, these below suggestions will help you to improve the quality of your research paper.

Title: The title clearly reflects the findings of the manuscript but it's not too long.

Abstract: In the abstract give some numbers reflecting your results. At the moment it's too general.

Some abbreviations are not explained. Key word: 6 key words are enough, so no need to reduce the key words.

Introduction: In the introduction you gave the importance to the China. How about the rest of the Word dry areas. Is this the common practice also there or something to suggest to other countries?Last paragraph of your introduction the aims or objectives is not clear and well explain please rewrite your aims or objectives. If you add some latest literature's will be much better if it available.

M&M: Material and methods are well described with valid statistical analysis and well explain procedure. Please add the procedure how you get soil moisture because soil moisture data is used to get soil water storage.Results: Data are present on the 9 Figures and 2 Tables.

Results are explained very well and clearly explain each tables and figures and much butter for general readers of LAND journal.

Discussion: The part of the manuscript, which should be re-structured with sub-title section, is discussion. I write three sub-titles please follow these sub-titles and re-structured your discussion part.4.1. Soil physical properties4.2. Soil nutrients characteristics4.3. Root morphological and physiological traits.

Minor editing of English language is required.

Author Response

Thank you for reviewing our manuscript! You are kind and responsible reviewers, and the suggestions you have given are all valuable and very helpful for revising and improving our paper. We are very grateful for that. We have studied your comments carefully and have made corrections; Many thanks to the Editor and the Reviewers for your time and thoughtful comments, many of which have been incorporated into the revised manuscript.

Below are our detailed responses (in BOLD type) to Editor and Reviewer’s comments, (the page and line numbers refer to our revised manuscript):

Reviewer

General comments:

There is a lot of interesting data in the manuscript on the Potential mechanism of optimal tillage layer structure for improving maize yield and enhancing root growth in Northeast China. While the experimental design is balanced/appropriate and detailed time courses of various antioxidant enzyme activities are presented  at different critical stages of development over 2 years) in addition to root growth and agronomic data (yield components). However, the below suggestions will help you to improve the quality of your research paper.

Response: respected reviewer, thank you very much for your encouragement and excellent suggestions.

Title: The title clearly reflects the findings of the manuscript but it's not too long.

Response: respected reviewer, thank you very much for your encouragement.

Abstract: In the abstract give some numbers reflecting your results. At the moment it's too general. Some abbreviations are not explained.

Response: Thank you for your advice, I rewrote and added some numbers that reflect my results, please see the abstract in the new revised version of my paper. I also explained the abbreviations of first-time use in the abstract.

Key word: 6 key words are enough, so no need to reduce the key words.
Response: Thank you for your advice.

Introduction: In the introduction you gave the importance to the China. How about the rest of the Word dry areas. Is this the common practice also there or something to suggest to other countries?
Response: Thank you for your advice, sir In the introduction section I mention semi-arid regions of China so my research can be applied in the rest of the world which belong to semi-arid regions or dry land farming systems.

Last paragraph of your introduction the aims or objectives is not clear and well explain please rewrite your aims or objectives. If you add some latest literature's will be much better if it available.

Response: Thank you for your advice, sir According to your suggestion I have rewritten my paper's aims and objectives please see in new revised paper. I also add the latest references in my new revised manuscript. 

M&M: Material and methods are well described with valid statistical analysis and well explain procedure. Please add the procedure how you get soil moisture because soil moisture data is used to get soil water storage.

Response: Thank you very much. Sir according to your advice I added the full-length procedure of soil moisture, Please see the new revise manuscript.

Results: Data are present on the 9 Figures and 2 Tables. Results are explained very well and clearly explain each tables and figures and much butter for general readers of LAND journal.

Response: respected reviewer, thank you very much for your encouragement.

Discussion: The part of the manuscript, which should be re-structured with sub-title section, is discussion. I write three sub-titles please follow these sub-titles and re-structured your discussion part.

4.1. Soil physical properties

4.2. Soil nutrients characteristics

4.3. Root morphological and physiological traits

Response: Thank you for your advice, Following your suggestion I restructured the whole discussion with three sub-title sections, please see in new revised manuscript.

Minor editing of English language required

Response: Thank you, Following your suggestion, I corrected all grammar mistakes according to your above suggestion, sir we have also sent our manuscript to a professional, native English-speaking Scientific Editor to improve the language and specifically to remove grammar mistakes.

Reviewer 2 Report

Moderate editing of English language required.

Author Response

Thank you for reviewing our manuscript! You are kind and responsible reviewers, and the suggestions you have given are all valuable and very helpful for revising and improving our paper. We are very grateful for that. We have studied your comments carefully and have made corrections; Many thanks to the Editor and the Reviewers for your time and thoughtful comments, many of which have been incorporated into the revised manuscript.

Below are our detailed responses (in BOLD type) to Editor and Reviewer’s comments, (the page and line numbers refer to our revised manuscript):

The manuscript has been well prepared, and the authors have conducted valuable research that can benefit maize farmers in China. The study investigated the Potential mechanism of optimal tillage layer structure for improving maize yield and enhancing root growth in Northeast China. The aim of this study was to evaluate the effect of different tillage structures soil physical and chemical properties, determine maize root morphological and Physiological characteristics under different tillage structures, study yield changes of different tillage layer structures and classify the relationship yield and oil physical and chemical properties, determine maize root morphological and Physiological characteristics to indicate the profitable tillage systems. Overall, the work is well reported and documented, and it is suitable for publication in the LAND Journal. However, there are several moderate flaws that need to be addressed:

Response: respected reviewer, thank you very much for your encouragement and excellent suggestions.

Introduction:

Clearly state the research hypothesis/question and the objectives of the study. This will help readers understand the importance of your study and the differences from previous studies.

 Response: respected reviewer, thank you very much for your encouragement.

Experimental Design:

Quantify both deep tillage and the given soil compaction in the soil bad to understand the dynamics of these practices and their potential interactions.

 Response: respected reviewer, thank you very much for your encouragement.

Data Analysis:

Specify whether the normality of the data was tested, especially for enzyme quantification in leaves.

Response: Thank you for your advice, Yes sir the normality of data is analysis and tested especially the enzyme or ROS quantification in soybean leaves.

Typos and Errors:

Correct "rain-fid" to "rain-fed" and similar mistakes throughout the manuscript.

Ensure consistency in terminology, such as removing unnecessary occurrences of "Under."

Response: Thank you for your advice, sir According to your above suggestion the correct rain-fid into rain-fed, also I remove the unnecessary occurrences of the “Under” word in my whole article, sir please see the new revise article.

Nitrogen Application:

Explain why nitrogen was not applied in a split application and discuss the potential risk of nutrient loss through leaching.

 Response: Thank you very much. Sir soybean is a nitrogen-fixing plant, so soybean just needs a starter dose of nitrogen, after that its fix the nitrogen with the help of root noodle, which is why we did not apply N in a split application.

Soil Water Content:

Provide information on whether the soil water content meters were calibrated for the specific site.

 Response: Thank you very much. Sir according to your advice I have added the full-length procedure of soil moisture, Please see the new revise manuscript.

The soil water content was calculated during 2016 and 2017 year. Moisture contents of the 0-60 cm soil layers at 10 cm intervals were recorded using a TDR meter (Time-Domain Reflectometry, Germany).

Language and Style:

Edit the manuscript for grammar and clarity, and consider having it reviewed by a native English speaker to improve the language quality.

Response: Thank you, Following your suggestion, I corrected all grammar mistakes according to your above suggestion, sir we have also sent our manuscript to a professional, native English-speaking Scientific Editor to improve the language and specifically to remove grammar mistakes.

Please cite this paper. Soil extracellular enzyme activities under long-term fertilization management in the croplands of China: a meta-analysis

Response: Thank you very much. Sir according to your advice I added the above article to my research article, sir really helped me a lot to improve my article, please see the new revised manuscript.

Miao, F., Li, Y., Song, C., Sindhu, J., Guofeng, Y., Qingping, Z. 2019. Soil extracellular enzyme activities under long-term fertilization management in the croplands of China: a meta-analysis. Nutr Cycl Agroecosyst, 114;125–138. https://doi.org/10.1007/s10705-019-09991-2

Conclusion:

Include a discussion of the impact of the study, particularly its contributions to grain yield under different treatments of deep tillage and the given soil compaction in the soil bad.

Response: Thank you, Following your suggestion, I included a discussion of the impact of the study, particularly its contributions to grain yield under different treatments of deep tillage and the given soil compaction in the soil bad, please see a new revised article.

Comments on the Quality of English Language

Moderate editing of English language required.

Response: Thank you, Following your suggestion, I corrected all grammar mistakes according to your above suggestion, sir we have also sent our manuscript to a professional, native English-speaking Scientific Editor to improve the language and specifically to remove grammar mistakes.
